# Anatomical Features of the Sphenoid Sinus and Their Clinical Significance in Transsphenoidal Accesses to the Pituitary Gland and Parasellar Region: A Systematic Review

**DOI:** 10.3390/diagnostics15243125

**Published:** 2025-12-08

**Authors:** Kristian Bechev, Antoaneta Fasova, Nina Yotova, Daniel Markov, Vladimir Aleksiev

**Affiliations:** 1Department of General and Clinical Pathology, Medical University of Plovdiv, 4002 Plovdiv, Bulgaria; daniel.markov@mu-plovdiv.bg; 2Neurological Surgery, Pulmed University Hospital, 4000 Plovdiv, Bulgaria; 3Department of Anatomy, Histology and Cytology, Faculty of Medicine, Medical University of Plovdiv, 4002 Plovdiv, Bulgaria; antoaneta.fasova@mu-plovdiv.bg (A.F.); ninayotova2004@yahoo.com (N.Y.); 4Department of Clinical Pathology, University Multidisciplinary Hospital for Active Treatment (UMHAT) “Pulmed”, 4002 Plovdiv, Bulgaria; 5Department of Thoracic Surgery, University Multidisciplinary Hospital for Active Treatment (UMHAT) “Kaspela”, 4002 Plovdiv, Bulgaria; vl_alex@abv.bg; 6Department of Cardiovascular Surgery, Medical University of Plovdiv, 4002 Plovdiv, Bulgaria

**Keywords:** sphenoid sinus, anatomical variations, transsphenoidal surgery, skull base, surgical complications

## Abstract

**Background:** The sphenoid sinus is essential for transsphenoidal surgical accesses to the sellar and parasellar regions because of its anatomic proximity to vital vascular and neurologic structures such as the internal carotid artery, optic nerve, and cavernous sinus. The high degree of morphological variability of the sphenoid sinus has a significant impact on surgical technique and the risk of intraoperative complications. Detailed knowledge of individual anatomy is therefore crucial for the safety and efficacy of transsphenoidal approaches. **Objectives:** This review aims to conduct a systematic analysis of the current scientific literature on anatomical variations in the sphenoid sinus and their clinical relevance in surgical interventions to the skull base. Special attention is paid to the influence of morphological features on surgical strategies to pathological processes in this area and postoperative outcomes. **Materials and Methods:** A systematic review of the literature was conducted according to PRISMA 2020 guidelines. The PubMed, Scopus, Web of Science, and Google Scholar databases were searched for the period March 2010 to March 2025. Keywords such as “sphenoid sinus”, “anatomical variations”, “transsphenoidal surgery” and “skull base” were used. Original studies, systematic reviews, and meta-analyses focused on the anatomy, pneumatization, and surgical significance of sphenoid sinus variations are included. Quality and relevance criteria for published material were considered in the selection of articles. **Results:** The most commonly identified anatomic variations included sellar and lateral pneumaticity, the presence of Onodi cells, multiple and deviated septa, and dehiscence of the posterior wall of the sphenoid sinus and prolapse into its cavity of the internal carotid artery. These variations are associated with an increased risk of intraoperative vascular injury, visual deficit, and postoperative liquorrhea. Accurate preoperative assessment by high-resolution computed axial tomography and magnetic resonance imaging, as well as the use of intraoperative neuronavigation, are critical to reduce surgical risk. **Conclusions:** Anatomic variations in the sphenoid sinus are an essential factor to consider when planning and performing transsphenoidal surgical accesses. An individualized approach based on detailed diagnostic imaging analysis and neuronavigation technologies contributes to a higher safety of the performed surgical interventions, a better radicality of tumor resection and more favorable postoperative outcomes.

## 1. Introduction

The sphenoid sinus (sinus sphenoidalis, SS) is a centrally located anatomical structure among the paranasal sinuses, characterized by high anatomical variability, which has important clinical significance in transsphenoidal surgical accesses to the pituitary gland and paracervical region [1,2,3,4,5]. Its close anatomical relationship to important structures such as the optic nerves, internal carotid artery and cavernous sinus requires a thorough knowledge of individual morphology to ensure safe and effective surgical access [3,4,5,6,7]. The degree and direction of sphenoid sinus pneumatization, the presence of septa, variations such as Onodi cells, and the absence of bony septa significantly influence surgical strategy and the risk of intraoperative complications [5,6,7]. Despite advances in surgical technique, the failure to thoroughly and in detail master these anatomical features remains a major risk factor for vascular trauma, visual impairment, and postoperative liquorrhea [1,2,3,4,5,6,7,8].

The development of high-resolution imaging—computed axial tomography (CAT) and magnetic resonance imaging (MRI)—allows detailed preoperative assessment of the sphenoid sinus and adjacent structures [9,10,11,12,13]. Incorporation of intraoperative neuronavigation techniques further improves safety in variable anatomy and significantly reduces the risk of intraoperative errors [10,11,12]. Despite the plethora of literature on the subject, a systematic analysis of the current literature on morphologic variations in the sphenoid sinus and their clinical significance remains necessary to establish a standardized approach in craniofacial surgery. This review aims to summarize and analyze the current scientific evidence on the topic.

## 2. Materials and Methods

This systematic literature review was conducted in accordance with the PRISMA 2020 (Preferred Reporting Items for Systematic Reviews and Meta-Analyses) guidelines from Appendix A. The aim was to identify, evaluate, and synthesize the available scientific evidence on anatomical variations in the sphenoid sinus and their clinical relevance in transsphenoidal surgical accesses to the pituitary fossa and parasellar region.

In the process of gathering information for the literature review, the PubMed, Scopus, Web of Science, and Google Scholar databases were searched for the period from March 2010 to March 2025. The search was performed using the following keywords alone or in combination: ‘sphenoid sinus’, ‘anatomical variations’, ‘transsphenoidal surgery’, ‘skull base surgery’, ‘pneumatization’, ‘onodi cell’, ‘complications’, ‘computed tomography- CT’, ‘magnetic resonance imaging- MRI’. Additional filters were used to limit the search to full-text, English-language articles. The results were reviewed and analyzed by each of the investigators, and appropriate inclusion and exclusion criteria for articles in the systematic review were validated.


*Inclusion Criteria:*
•Original clinical studies, systematic reviews and meta-analyses.•Publications in English with full text available, peer-reviewed and refereed articles of high scientific value.•Studies examining the anatomy, morphological variations and surgical significance of the sphenoid sinus.•Articles using diagnostic imaging modalities (CT and MRI).•Persons over 18 years of age with completed pneumatization of the sphenoid sinus



*Exclusion criteria:*
•Articles without a clearly defined methodology.•Papers and publications focusing on other anatomical regions.•Repeated publications or studies with low-quality literature data.•Publications in languages other than English.•Unreviewed and unrefereed articles.•Persons of childhood age whose sinus pneumatization is not yet complete.


The initial number of articles identified was 178. After reviewing the latter, duplicate articles (*n* = 28) were removed, leaving 150 publications for title and abstract review. Of these, 59 were excluded from the study due to a lack of relevance or focus on other anatomical regions of the head and face. The full texts of 91 articles were assessed by each investigator on the team for compliance with the study inclusion criteria. After which, according to the inclusion and exclusion criteria for research articles, 53 full-text publications remained eligible to be included in our study. Data regarding authorship, year of publication, imaging methods, anatomic variations described, clinical implications, and surgical complications were extracted from the included articles. The information was systematized and analyzed quantitatively and qualitatively. Clinical and imaging-referenced studies predominated, in which computed axial tomography (CAT) was used as the main diagnostic tool to assess sinus morphology and identify anatomical variations at risk, such as multiple septa, Onodi cells and dehiscence of the posterior wall of the sphenoid sinus and prolapsing internal carotid artery.

Risk of bias was assessed using adapted checklists for selection, interpretation, and publication bias. The Newcastle–Ottawa Scale (NOS) was applied for observational studies and the AMSTAR 2 tool for systematic reviews. Two independent reviewers performed the primary review of titles and abstracts. In case of disagreement, a third reviewer adjudicated the case. The final number of included articles was determined after critical analysis of the full text by the co-authors. We present a PRISMA 2020 diagram of the publication selection process (Figure 1).

## 3. Results

After applying the selection criteria, a total of 53 scientific publications were included. These included: 29 original clinical studies, 12 systematic reviews, 5 meta-analyses and 7 small series with surgical observations.

The articles studied included 8000 patients undergoing imaging and/or transsphenoidal surgical interventions. Computed axial tomography (CAT) and high-resolution magnetic resonance imaging (MRI) were the most commonly used methods to evaluate anatomy, and three-dimensional reconstruction of anatomic data was used in 72% of the studies.

Analysis of the most commonly described anatomic variations in the sphenoid sinus included in the study showed the following distribution:•Sellar type pneumatization.•Presellar type pneumatization.•Conchal type pneumatization (no or minimal pneumatization).•Lateral pneumatization of the greater wings of the sphenoid.•Presence of Onodi cells.•Dehiscent internal carotid artery (absence of bony barrier in front of the internal carotid artery).•Multiple septa in the sphenoid sinus, often diverted to the ICA or optic canal.

Graphically, the anatomical variations in SS can be represented as follows (Figure 2).

The figure illustrates the frequency of the main anatomical variations in the sphenoid sinus identified by imaging and anatomical studies. The most common type of pneumatization is sellar, found in approximately 72% of patients, which corresponds to the clinically preferred transsphenoidal approach because of the direct anatomic relationship to the sella turcica. Presellar and conchal types are less common (12% and 4%, respectively), in which the surgical approach is more difficult and requires caution during bony resection. Lateral pneumatization (20%) increases the risk of entering the cavernous sinus or near the internal carotid artery. Variations such as the presence of Onodi cells (11%) and dehiscent ICA (8%) are associated with an increased risk of visual and vascular complications, especially in the absence of preoperative identification. Multiple septa (28%) are also of clinical significance as they can deviate and fixate to critical structures, making navigation difficult during transsphenoidal interventions. These data highlight the need for individualized preoperative imaging assessment to identify morphologic variations in SS in order to reduce intraoperative risks and improve surgical outcomes.

Based on the summarized literature data, the anatomic variations occurring most frequently can be presented in tabular form according to the type of variation, its description, and their percentage distribution (Table 1).

Multiple studies have reported an increased risk of intraoperative complications in the presence of certain variations:•Dehiscent ICA increases the risk of massive hemorrhage during manipulation in the sphenoid sinus.•Onodi cells can be misinterpreted as part of SS and lead to optic nerve damage.•Multiple septa attached to the ICA or optic canal make navigation difficult and increase the risk of trauma.•Lateral pneumaticity can impede access and increase the risk of liquorrhea and vascular complications.

All included studies emphasize the role of high-resolution imaging using CTA in preoperative mapping of the sphenoid sinus. CT with thin sections (≤1 mm) in axial and coronal projections is considered the “gold standard”. MRI complements the evaluation by visualizing soft tissue structures, especially in the presence of tumor invasion to the cavernous sinus or optic nerve. Three-dimensional reconstructions significantly improve preoperative orientation and allow more precise surgical planning in cases with complex anatomical configurations.

## 4. Discussion

The sphenoid sinus (SS) is an anatomical structure located at the base of the sphenoid bone of the human skull that begins its development relatively late compared to the other paranasal sinuses. Its origin is associated with the posterior ethmoidal cells, which during the third month of intrauterine development begin the process of pneumatization of the cuneiform bone (os sphenoidale) [1,2,3,4,5,6,7,8,9,10]. The process of complete pneumatization is usually complete by late adolescence, around 18–20 years of age [3,4,5,14,15,16,17,18]. Individual variations in sinus development are explained by differences in the rate of mucosal invagination and the degree of bone resorption [4,5,6,7,8,9,10,11,12,13,14,19,20]. At the molecular level, the development of the paranasal sinuses, including the sphenoid sinus, is controlled by a complex regulation of genetic factors such as MSX1, MSX2, and PAX9, which are involved in osteogenesis and skull base remodeling [5,6,7,8,9,10,11,12,13,19,20]. The embryonic origin of the sphenoid sinus explains why its walls have an immediate anatomical relationship with important structures such as the optic nerve, the internal carotid artery, and the cavernous sinus (containing cranial nerves III, IV, V1, V2, and VI) [6,16,17,18]. The degree of pneumatization varies individually and is determined by embryonic development, genetic factors, and environmental factors. As pneumaticity increases, the volume of the SS increases, with its walls becoming thin and immediately adjacent to important anatomical structures. This anatomical proximity has a determinant role in the safety of transsphenoidal surgical interventions, as variations in sinus development can affect operative risk [5,6,9].

The morphological characterization of the sphenoid sinus shows considerable individual variability in terms of its volume, shape and degree of pneumatization. These features are essential for clinical practice as they directly influence access to the sellar and parsellar regions during surgical interventions [1,14,15,16,17,18,20,21,22,23,24,25,26,27,28,29,30]. Pneumatization of the sphenoid sinus usually starts from the posterior portion of the ethmoidal labyrinth system and gradually extends into the body of the cuneiform bone. Depending on the distribution of the air space relative to the sella turcica, three main types of pneumatization are described:•Sellar type: In this most common variant (65–80% of cases), pneumatization passes behind the dorsum sellae and involves the sellar fossa. In the pneumatication variant, the thin bony septum between the sphenoid sinus and the turk’s saddle creates the best conditions for direct surgical access to the pituitary gland [2,3,13,14,15,16,17,18,19,20,21,22,23]. This option is preferred for transsphenoidal interventions because of the shorter and safer surgical path.•Presellar type: occurs in about 10–15% of the population. In this variant, the air cavity is restricted in front of the sella turcica, leaving a thick layer of bony tissue between the sinus and the pituitary gland [4,14,15,16,17,18,21,22,23,24,25,26,27,28]. This pneumatization makes surgical access difficult, necessitates more extensive bone resection, and increases the risk of intraoperative complications.•Conchal type: represents the rarest variant of pneumatization (2–5%) in which the sphenoid sinus is severely reduced or almost completely absent. In these cases, the body of the cuneiform bone is filled with dense bony tissue without a significant air cavity [5,8,9,18,21,22,23,24,25,26,27,28]. Operative access to the acetabulum in such cases is extremely difficult and associated with an increased risk of complications [17,18,21,22,23,24,25,26,27,28,29].

In addition to the basic types, a number of authors describe additional variations in SS pneumatization. These are: lateral pneumatization, in which air cavities are seen to spread to the greater wings of the cuneiform bone, the pterygoid process, or the cavernous sinus. The presence of lateral pneumatosis makes orientation difficult during surgical interventions and increases the risk of liquorrhea and damage to adjacent neovascular structures [6,25,26,27,28,29,30,31,32]. Clavicular pneumatization results in expansion of the sphenoid sinus toward the clivus. This variant has implications for interventions to the posterior cranial fossa and brainstem. Other morphological features of the sinus include more:•Presence of interseptal septations (septa), which may be centrally located, deviated, or multiple.•Atypical septal configurations that are sometimes attached to the wall of the internal carotid artery or to the optic canal.

These details are of primary importance in surgical planning, as misinterpretation of septal structures can lead to surgeon deception and subsequent intraoperative complications [7,9,19,22,23,28]. Observations have shown that the morphological variability of SS is influenced by factors such as age, gender, and ethnicity. For example, Asian populations demonstrate a higher incidence of presellar and conchal pneumatosis compared to European populations [8,10,11,12]. Understanding these anatomical features is paramount to safely performing transsphenoidal surgical procedures and minimizing the risk of neovascular complications.

The sphenoid sinus is known for its morphologic variability, which manifests in different anatomic variations. Knowledge of the latter is important for the safe performance of transsphenoidal surgical accesses, especially in pituitary and paracervical surgeries [1,2,3,14,20,27,29].

*Onodi cells*.

Onodi cells represent a posterolaterally located ethmoidal cell that may extend over the sphenoid sinus or fuse with it. The characteristic feature of these cells is that the optic nerve, and sometimes the internal carotid artery, may pass into their wall [2,3,4,5,6,7,8,9,10,11,12,13,14,15,16,17,18,19,20,21,22,23,24,25,26,27,28,29,30,31]. The incidence of Onodi cells varies between 8% and 14% in different populations. The clinical significance of this variation is very large, as misidentification of an Onodi cell can lead to serious complications such as optic nerve damage and subsequent loss of vision on the side of the affected optic nerve [3,4,5,6,7,8,9,10,11,12,13,14,15,16,17,18,19,20,21,22,23,24,25,26]. Identification of Onodi cells by high-resolution CTA or MRI preoperatively is mandatory to minimize the risk of neurological damage during transsphenoidal interventions [3,4,5,6,7,8,9,10,11,12,13,14,15,16,17,18,19,20,21,22,23,24,25,26,27,28,29,30,31,32].


*Absence of bony barrier in front of the internal carotid artery (Dehiscent ICA).*


In some individuals, the bony septum separating the SS from the ICA is thinned or completely absent. Deficiency of the bony septum (dehiscent ICA) is seen in about 5–10% of cases [4,12,16,23]. This anatomic feature poses a serious risk of ICA rupture and subsequent massive intraoperative hemorrhage, which can be fatal. Therefore, careful evaluation of the bony structures surrounding the ICA with high-resolution CT imaging is an essential element of preoperative planning [1,2,3,4,5,6,7,8,9,10,11,12,13,14,15,16,17,18,19,20,21,22,23,24,25,26,27,28,29,30,31,32,33,34,35].


*Multiple and deviant septa.*


SS septa are bony septa that may be single, multiple, or deviated. They divide the sinus cavity into different compartments and vary considerably between populations as well as within populations. About 20–35% of patients have multiple septa in the sphenoid sinus [5,6,18]. Particular care is required when septa are attached to critical structures such as the ICA or optic canal. During surgical interventions, removal of such septa without neuronavigation can lead to surgeon deception and accompanying vascular or neurological complications [1,2,3,4,5,6,7,8,9,10,11,12,13,14,15,16,17,18,19,20,21,22,23,24,25,26,27,28,29,30,31,32,33,34].


*Lateral Pneumatization.*


Lateral pneumatization represents extension of the sinus sphenoidalis to the structures of the greater wing of the cuneiform bone, the processus pterygoideus, or the cavernous sinus [6,10,11,12,13,14,15,16,17,18,19,20,21,22,23,24,25,26,27,28,29,30,31,32,33,34,35,36,37,38]. The clinical significance of lateral pneumatization is associated with an increased risk of disruption of the bony skull base, which can lead to intraoperative liquorrhea and postoperative infections such as meningitis [10,28,29,32].


*Variations in the localization of the optic canal and ICA.*


An additional feature is created by anatomical variations in which the optic canal or ICA is partially protruded into the sphenoid sinus or completely covered only by mucosa [7,8,9,13,25]. These variations increase the risk of direct damage during instrumental manipulation during transsphenoidal access. Preoperative three-dimensional mapping of these structures is essential for the safety of the intervention [7,8,9,10,11,12,13,14,15,16,17,18,19,20,21,22,23,24,25,26,27,28,29,30,31,32,33,34,35,36].

Preoperative imaging occupies a central place in the planning of transsphenoidal surgical interventions. Due to the morphological variability of the sphenoid sinus and its anatomical relationship with important neovascular structures, a detailed evaluation of the individual anatomy is absolutely essential [1,2,3,4,5,6,7,8,9,10,11,12,13,14,15,16,17,18,19,20,21,22,23,24,25,26,27,28,29,30,31,32,33,34,35,36,37,38,39,40]. Classical CTA is the “gold standard” for preoperative evaluation of the bony structures of the SS and its surrounding anatomic structures. It is recommended to use sections ≤1 mm thick; scanning in axial, coronal, and sagittal planes; and bone window reconstructions for better visualization of septa, absence of bone wall, and protrusions of the ICA and optic nerve into the SS lumen [1,2,3,4,5,6,7,8,9,10,11,12,13,14,15,16,17,18,19,20,21,22,23,24,25,26,27,28,29,30,31,32,33,34,35,36,37,38,39,40]. In addition, CTA allows precise assessment of: the type of pneumatization of the sinus (sellar, presellar, conchal); the presence of Onodi cells; lateral spread of pneumatization to the cavernous sinus and the wings of the cuneiform bone [1,2,3,4,5,6,7,8,9,10,11,12,13,14,15,16,17,18,19,20,21,22,23,24,25,26,27].

MRI complements the CAT scan diagnosis by providing a detailed assessment of soft tissue structures that are not as visible on a CAT scan. These are: pituitary gland (size, shape, presence of adenoma), optic chiasm and optic nerves, cavernous sinuses and their invasion by tumor processes, the relation of the lesion to important vascular and neural structures. The T1-sequence with and without contrast and the T2-sequence allow detailed differentiation between different pathological processes, such as micro- and macroadenomas, meningiomas, craniopharyngiomas and other neoplasias of the parasellar region [1,2,3,4,5,6,7,8,9,10,11,12,13,14,15,16,17,18,19,20,21,22,23,24,25,26,27,28,29,30,31]. In cases of large adenomas, MRI is indispensable to determine the degree of suprasellar or lateral spread, which determines the choice of surgical approach [30,31,32,33,34,35,36,37,38,39,40,41].

In the last decade, Cone-Beam CT (CBCT) has established itself as an extremely precise technique for the evaluation of SS with minimal radiation exposure. CBCT provides: three-dimensional (3D) reconstructions of anatomical structures, excellent spatial resolution of bone details, and reduced radiation dose compared to conventional CT [5,17,18,23,27]. CBCT is particularly useful in patients with previous surgeries in which the anatomy has been altered, and in the evaluation of complex anatomic variations such as lateral pneumatization or atypical septa [5,6,7,8,9,10,11,12,13,19,20]. In current practice, preoperative CT and MRI data are often used to create virtual 3D models of the SS and surrounding structures. These models allow: precise preoperative simulation of the surgical approach, identification of important points for neuronavigation, and optimization of the surgical strategy according to the individual anatomy [6,7,8,9,10,11,12,13,14,15,16,17,18,19,20,21,22,23,24,25,26,27,28,29,30,31,32,33,34,35,36,37]. The integration of virtual planning with intraoperative navigation systems reduces the incidence of intraoperative complications and increases the safety of interventions.


*Surgical techniques, intraoperative risks and complications of transsphenoidal surgery.*


The transsphenoidal approach to the pituitary gland represents one of the significant innovations in neurosurgery, and its development has passed through several milestones from the late 19th century to modern minimally invasive surgery. The first attempts at surgical access to the pituitary gland were made via the transmaxillary and transnasal routes, but it was not until the introduction of the microscope in the 1960s that the technique gained safety and precision, allowing less invasive manipulation of structures along the skull base [1,2,3,4,5,6,7,8,9,10,11,12,13,14,15,16,17,18,19,20,21,22,23,24,25,26,27,28,29,30,31,32,33,34,35,36,37,38,39,40,41,42]. In the last two decades, endoscopic transnasal surgery has revolutionized its approach to the SS and pituitary region. Endoscopic techniques offer a wider field of view, better visualization of anatomic details, and the ability to work in narrow anatomic spaces by accessing through one nostril or both. Angled endoscopes (30°, 45°, 70°) allow the surgeon to visualize laterally located lesions that traditionally would be difficult to access with a microscopic approach [2,3,4,5,6,7,8,9,10,11,12,13,14,15,16,17,18,19,20,21,22,23,24,25,26,27,28,29,30,31,32,33,34,35,36,37,38,39,40].

The microscopic transsphenoidal approach, although classical, still finds application in certain cases, especially in small well-encapsulated adenomas, where the direct line of access and stable triangulation of the instruments offer advantages. However, limitations in the field of view and lack of lateral visualization make the microscopic technique more risky in invasive tumors with cavernous or suprasellar invasion [18,26,27,28,30,36,37,38,39,40,41,42,43]. Combined endoscopic–microscopic techniques find application in complex cases where both a wide field of view and high precision manipulation are required. In giant adenomas with retrosellar spread or in recurrent lesions, this approach combines the advantages of both techniques. The variety of anatomic variations in the sphenoid sinus requires an individualized choice of surgical strategy. Sellar pneumatization usually allows easy access, whereas presellar and conchal types often require additional bony resection. The presence of Onodi cells and the defect in the bony septum between the SS and the internal carotid artery increase the risk of vascular complications with subsequent neurological deficit, necessitating the use of intraoperative navigation and careful microdissection [4,5,6,7,8,9,10,11,12,13,14,15,16,17,18,19,20,21,22,23,24,25,26,27,28,29,30,31,32,33,34,35,36,37,38,39,40,41].

Navigation technologies based on preoperative CT and MRI mapping have become standard in high-risk transsphenoidal surgical interventions. Intraoperative navigation systems provide real-time localization of surgical instruments relative to anatomic structures, reducing the risk of vascular injury by up to 40% according to recent meta-analyses [5,11,23,31,38,39,40,41,42,43,44,45,46,47]. Navigation systems allow accurate localization of anatomical landmarks, avoidance of vascular and neurological damage, and minimization of resection of healthy bone tissue. Studies have shown that the use of navigation reduces complications by 30 to 50% in transsphenoidal surgical interventions [12,13,30,31,32,33,34,35,36,37,38,39,40,41,42,43,44,45,46,47,48]. Virtual surgical planning using 3D modeling further aids planning for complex anatomic configurations and tumor lesions with atypical spread. The most serious intraoperative risks of transsphenoidal interventions include vascular complications, such as internal carotid artery rupture, which, although rare (<1%), has an extremely high mortality rate when it occurs. Clinical cases of intraoperative ICA rupture have shown that survival is dependent on prompt diagnosis and action, such as direct compression, use of balloon occlusion, or stent-assisted embolization. In rare situations, it is necessary to perform endovascular occlusion of the artery, which, however, indicates a high risk of cerebral infarction [2,3,4,5,6,7,8,9,10,11,12,13,14,15,16,17,18,19,20,21,22,23,24,25,26,27,28,29,30,31,32,33,34,35,36,37,38,39,40]. Damage to the optic nerve during manipulation in the Onodi cell area is the second-most significant risk and can lead to blindness. Other common complications are liquorrhea due to disruption of the integrity of the bony skull base, dural sheath lesions, and subsequent infections such as meningitis [6,7,8,9,10,11,12,13,14,15,16,17,18,19,20,21,22,23,24,25,26,27,28,29,30,31,32,33,45,46,47]. Retrospective studies and meta-analyses covering more than 5000 transsphenoidal surgeries have shown that the incidence of liquorrhea with endoscopic technique is about 3–5%, whereas with microscopic technique it is between 2% and 4%, but endoscopic surgery achieves significantly better results in resection of invasive macroadenomas and in controlling bleeding [7,8,18,25,30,31,32,33,34,35,36,37,38,39,40,41,42,43,44]. Strategies to minimize intraoperative risks include rigorous preoperative high-resolution imaging evaluation, use of navigation systems, sparing dissection in the presence of deformed or thinned bone structures, and provision of vascular reconstruction. When a compromised bony ICA sheath is suspected, the use of additional endoscopic intraoperative Doppler sonography for localization of the vascular lumen is recommended [8,9,10,11,12,13,14,15,16,17,18,19,20,21,22,23,24,25,26,27,28,29,30,31,32,33,34,35,36]. Recent years have also seen the integration of intraoperative MRI (iMRI) to assess the extent of tumor resection during intervention. This allows immediate correction of incomplete tumor resection and reduces the need for revision surgeries. Modern transsphenoidal surgery represents a balanced synthesis between technological advances and anatomical knowledge, where success depends not only on the technique chosen but also on thorough preoperative planning and a precise intraoperative strategy adapted to the individual anatomy of each patient [14,15,16,17,18,20,21,22,23,24,25,26,27,28,29,30].

Transsphenoidal surgical interventions are generally considered safe procedures with low rates of severe complications, especially when performed by experienced teams and with strict preoperative planning. However, rare but potentially fatal complications exist in clinical practice that require special attention. The development of postoperative hypopituitarism due to compromise of normal pituitary tissue during resection of macroadenomas has been described in the literature. Studies have shown that the incidence of new-onset total hypopituitarism after transsphenoidal surgery is between 2% and 5%, with risk factors including large tumor size, suprasellar spread, and the need for aggressive resection [3,4,6,30,31,32,33,34,35,36,37,38,39,40,41,42,43,44,45]. Isolated cases of optic neuropathy after transsphenoidal interventions have also been reported. The pathophysiologic mechanism most commonly involves direct trauma to the optic nerve, compression by hematoma, or ischemia associated with manipulation near the optic chiasm. Early diagnosis by postoperative MRI scanning and removal of compressive factors are fundamental manipulations for restoration of visual function [39,40,41,42,43,44,45,46,47,48].

Liquorrhoea, although a more common complication, can in rare cases lead to the development of meningitis, especially if not promptly diagnosed and treated. Standard surgical practice recommends the use of a vascularized nasoseptal flap to close skull base defects in the presence of intraoperative liquorrhea, and this significantly reduces the incidence of postoperative infections [5,10,18,20,21,22,23,24,25,26,27,28,29,30,31,32,33,34,35,36,37,38,39,40,41,42,43,44,45,46,47]. Rare complications described in the literature include cases of ectopic adenomas in the sphenoid sinus, which may initially be mistaken for an invasive macroadenoma. In these cases, differential diagnosis by detailed MRI and histopathological evaluation is crucial for proper therapeutic management [6,23,24,25,26,27,28,29,30,31,32,33,34,35,36,37,42,43,44,45,46,47]. Of note is the occurrence of late complications such as sphenoid mucocele, which can develop years after transsphenoidal interventions, especially with incomplete filling of the sphenoid sinus and impaired mucosal drainage. Obstruction of the SS can lead to compression of surrounding structures and requires surgical revision [7,26,27,29,30,31,32,33,34,35,46,47,48,49]. Despite the rarity of these complications, their understanding and anticipation is mandatory for any neurosurgeon performing transsphenoidal accesses.


*Implications for future research in transsphenoidal surgery.*


Despite significant advances in the study of the anatomic features of the sphenoid sinus and transsphenoidal surgical approaches, there are a number of limitations in the available literature that must be considered when interpreting published results and formulating clinical guidelines. One of the main limitations is related to the heterogeneity of the methodologies used in the different studies. The lack of uniform standardization in the description of pneumatic types, anatomical variations and surgical complications makes it difficult to directly compare results between different study groups. In some studies, Onodi cells were defined based on topographic features relative to the optic nerve, whereas in others they were defined relative to the internal carotid artery, resulting in different reporting rates [1,11,14,18,21,22,23,24,25,26,27,36,37,38,39,40,41,42,43,44,45,46,47]. Another significant limitation is the dominance of retrospective studies based on data from single centers. Such studies carry risk because patient populations and surgical techniques vary widely between neurosurgical centers. The lack of large, prospective multicenter studies limits the ability to draw universal conclusions regarding the frequency of anatomic variations, the risk of complications, and the effectiveness of different surgical approaches [2,3,4,5,6,7,8,9,10,11,12,13,14,15,16,17,18,19,20,21,22,23,24,25,26,27,28,29,30,31,32,33,34,35,36,37,38,39,40,41,42,43,44,45,47,49,50,51,52]. Preoperative imaging also suffers from variability in the techniques and protocols used [3,4,5,6,7,8,9,10,11,12,13,14,15,16,17,18,19,20,21,22,23,24,25,26,27,28,29,30,31,32,33,34,35,36,37,38,39,40,41,42,43,44,45,46,47,48]. With regard to surgical techniques, comparative studies between microscopic and endoscopic transsphenoidal surgery are often subject to interpretation, as the choice of technique is determined not only by the characteristics of the lesion but also by the individual surgeon’s experience and preference [4,5,6,7,8,9,10,11,12,13,14,15,16,17,18,19,20,21,22,23,24,25,26,30,35,36,41,42,43,44,45,45,46,52]. It should also be noted that most published series include predominantly patients with pituitary adenomas, whereas data on rarer lesions such as craniopharyngiomas, meningiomas, or ectopic sphenoid sinus tumors remain limited. This creates a knowledge gap regarding the optimal surgical approach for such pathology [5,10,11,14,15,16,17,18,19,20,21,22,23,33,34,35,36,37,38].

## 5. Conclusions

The sphenoid sinus is an anatomical structure with a key role in allowing safe and efficient transsphenoidal access to the pituitary gland and paracervical region. Its morphological variability, manifested in different degrees of pneumatization, the presence of anatomical variations such as Onodi cells, the absence of a bony wall in front of the ICA and multiple septa, creates specific challenges for surgical practice. Detailed knowledge of the anatomy of the sphenoid sinus is essential for preoperative planning. High-resolution CBCT and MRI are indispensable for detailed assessment of individual anatomical features; technologies such as CBCT, 3D reconstructions, and intraoperative neuronavigation greatly improve the safety of surgical interventions. Modern transsphenoidal surgery is the result of an evolution of techniques from classical microscopic approaches to minimally invasive endoscopic interventions that provide better visualization, lower morbidity, and higher tumor resection rates. The choice of technique and surgical strategy should always be individualized, based on a detailed preoperative assessment of the patient’s anatomic variations and characteristics.

## Figures and Tables

**Figure 1 diagnostics-15-03125-f001:**
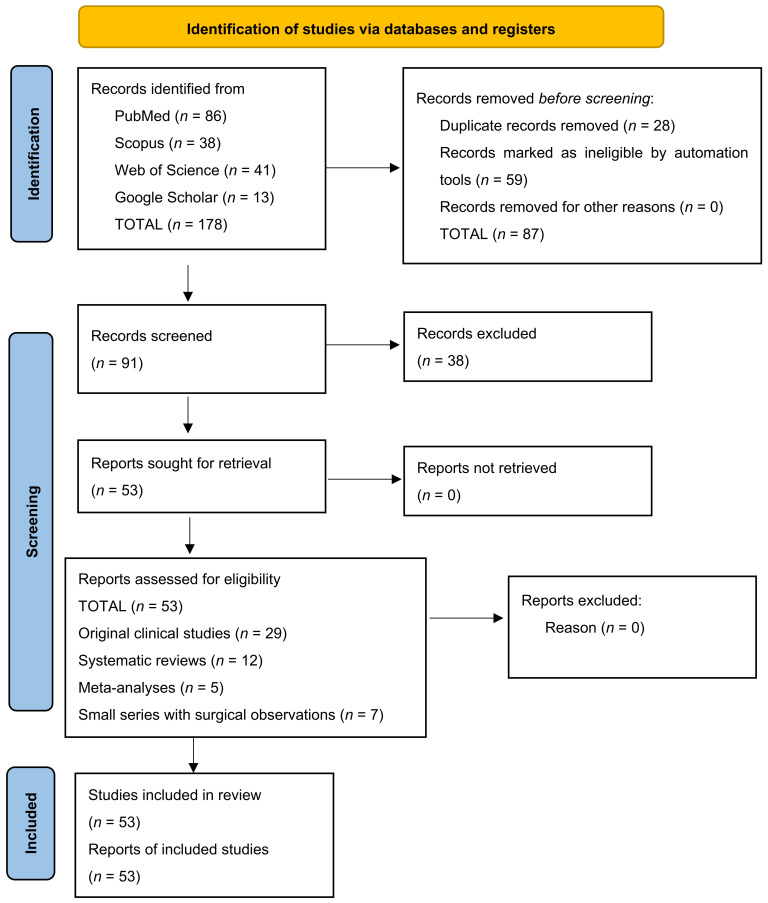
PRISMA 2020 diagram of the publication selection process. The diagram illustrates the process of systematic search, selection and inclusion of scientific publications in accordance with PRISMA 2020 guidelines. A total of 178 publications were identified through electronic database searches (PubMed, Scopus, Web of Science and Google Scholar).

**Figure 2 diagnostics-15-03125-f002:**
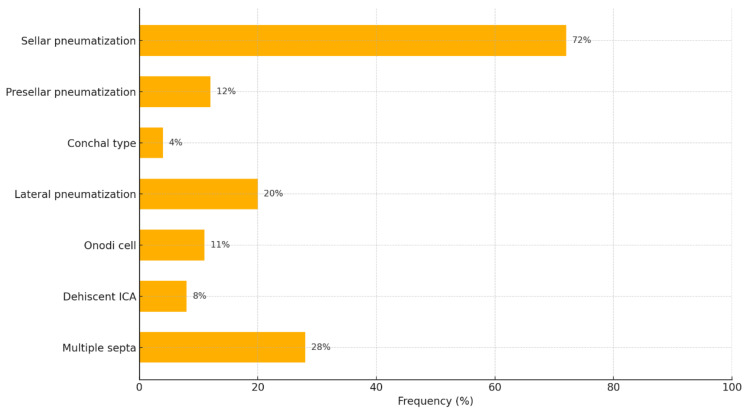
Frequency of major anatomical variations in the sphenoid sinus expressed as a percentage, including a total of 800 patients.

**Table 1 diagnostics-15-03125-t001:** Distribution of anatomical variation by type, description and percentage. (Some sources note considerable ethnic and gender variation in the prevalence of variation.)

Variation Type	Description	Average Frequency (%)
**Sellar pneumatization**	Pneumatization reaching the sella turcica	65–80%
**Presellar type**	Pneumatisation limited before sella	10–15%
**Conchal type**	Minimal or missing pneumatisation	2–5%
**Lateral recess extension**	Pneumatization to the wing of the sphenoid bone	15–25%
**Dehiscent internal carotid artery**	Lack of bony barrier between artery and sinus	5–10%
**Onodi cell**	Ethmoid cage entering the sphenoid sinus	8–14%
**Multiple intersinus septa**	Two or more septa diverted to the carotid or optic canal	20–35%

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
