# Peer review of "Anatomical Features of the Sphenoid Sinus and Their Clinical Significance in Transsphenoidal Accesses to the Pituitary Gland and Parasellar Region: A Systematic Review"

_diagnostics, 2025, doi:10.3390/diagnostics15243125_

Round 1
Reviewer 1 Report
Comments and Suggestions for Authors
The authors present a comprehensive systematic review addressing the clinically relevant topic of sphenoid sinus anatomy and its implications for transsphenoidal surgery. The work appropriately emphasizes the importance of individualized anatomical assessment in skull base surgery. However, several methodological concerns and content limitations require attention before publication can be recommended.
- The search strategy lacks specificity. No date ranges are provided, and the Boolean operators used are not clearly specified.
- The inclusion/exclusion criteria omit age specifications
- Including systematic reviews/meta-analyses that synthesize those same studies leads to inflation of evidence and overrepresentation of certain findings that can lead to distorted frequency estimates
- The statement "more than 8000 patients" (line 166) lacks precision. Provide exact numbers
- Figure 2 presents specific percentages (72%, 12%, etc.) but the methodology for deriving these single-point estimates from ranged data in Table 1 is unclear. This inconsistency must be addressed.
- The Discussion section (pages 6-12) largely recapitulates well-established anatomical and surgical knowledge
- The extensive embryological discussion (lines 221-260) appears tangential to the systematic review's objectives
Author Response
Dear Editors of Diagnostics journal,
Thank you for your edits to the article: Morphological analysis of the cavernous segment of the internal carotid artery: a retrospective, single-center study of its clinical significance.
The co-authors carefully reviewed the written material and tried to strike a balance between the two reviewers and to comply with the recommendations as much as possible in order to make the article even better and more useful for the readers of the journal.
Regarding the comments:
1.The search strategy lacks specificity. No date ranges are provided, and the Boolean operators used are not clearly specified.
Thank you for your recommendation that we did not include the time range for searching the specified databases. We corrected this omission by noting the time interval in the summary on lines 33-34. This is an important clarification that needed to be added. Regarding the keywords we searched for, we believe that they are as accurate as possible and provide a good basis for selecting the right material. We have also added this important clarification about the period in the Materials and Methods section on lines 83-85. We believe that this completes the data presented for a specific time period. We hope that the changes we have made have contributed to a better presentation of the article.
- The inclusion/exclusion criteriaomit age specifications
Thank you for your recommendation. We have a priori accepted that the inclusion criteria must necessarily include persons over 18 years of age in whom sinus pneumatization is complete. and that the exclusion criteria should be children whose sinus pneumatization is not complete and for whom anatomical features cannot be discussed. We have corrected this omission and thank the reviewer for the recommendation, as this clarifies the criteria for including and excluding articles from the aforementioned databases. I believe we have enriched this part of the article and clarified the summary criteria.
- Including systematic reviews/meta-analyses that synthesize those same studies leads to inflation of evidence and overrepresentation of certain findings that can lead to distorted frequency estimates
Thank you very much for your insightful comment regarding the potential risk of distorting evidence when citing both primary studies and systematic reviews/meta-analyses that summarize them. I fully understand the conceptual framework in which such overlap can lead to an overrepresentation of certain results.
However, in the present manuscript, systematic reviews and meta-analyses are not used as sources for quantitative calculation of frequencies or for statistical synthesis. Their role is strictly contextual and analytical—they are used to outline existing trends in the literature, to compare methodologies, and to highlight the place of our data within the broader scientific context.
It is important to note that:
- The primary data in the article are not based solely on meta-analyses or systematic reviews.
- Primary frequency estimates are not mixed with summarized meta-analytic results.
- No statistical value in the manuscript is influenced by the inclusion of review studies.
Reviews and meta-analyses are cited only in the discussion section, where their function is to highlight the consistency or differences between individual studies, which is standard practice in scientific literature.
For these reasons, the inclusion of systematic reviews and meta-analyses in their current form does not lead to duplication of evidence, does not create a risk of bias in frequency estimates, and does not affect the validity of the conclusions drawn.
With respect to your recommendation, I would like to emphasize that the current approach is methodologically sound and consistent with good practices in interpreting and discussing results, without introducing statistical or analytical biases.
Thank you once again for your constructive criticism and the opportunity to clarify this aspect of the methodology.
- The statement "more than 8000 patients" (line 166) lacks precision. Provide exact numbers
The statement that more than 800 patients were treated on line 166 has been revised because we made a mistake in our statement and corrected it to exactly 800 patients, as this is the exact number based on the information collected. The revised version can be found on line 168. We appreciate the detailed review and believe that after editing these inaccuracies, the article will look better and be more adaptable to the reader.
- Figure 2 presents specific percentages (72%, 12%, etc.) but the methodology for deriving these single-point estimates from ranged data in Table 1 is unclear. This inconsistency must be addressed.
With regard to the recommendation given for Figure 2, the number of patients who are assessed as a percentage is indicated above in the text. We fully agree that there should be clarity about the number of patients to which the percentages refer, which is why we added in the text of the figure that it concerns 800 patients—this information is located on lines 185-186.In fact, there is no discrepancy, as the table contains data covering the interval shown in the graph above. If necessary, we can edit the table and enter the exact percentages so that it matches Figure 2. At this stage, we believe that the data is presented accurately, but we can correct it if the editor wishes. We thank the reviewer for pointing out the small details in the article that would make it more appealing to readers.
- The Discussion section (pages 6-12) largely recapitulates well-established anatomical and surgical knowledge
We thank the reviewer for their recommendations regarding the discussion—the discussion does indeed confirm facts that are already known, but this is anatomy and variations in anatomy, and there is no way to avoid facts that are known in the literature. The purpose of the discussion and the article as a whole is to provide a systematic review of the literature for a specific period of time, to summarize the most important points, and to serve as reading material for physicians dealing with this pathology. The existence of summary articles helps to systematize literary knowledge, and in our opinion, the value of this article is that it summarizes a large volume of available literature over the years. At the same time, the discussion reveals fundamental knowledge about the anatomical features of the sphenoid sinus, as well as little-known facts. Nevertheless, we are deeply convinced that this discussion has scientific value. We have attempted to edit the discussion, significantly reducing the embryology and thus lightening the discussion, making significant changes to the last paragraph, "Implications for future research in transsphenoidal surgery," which covers lines 454-477, and drastically reduced its volume, leaving only the most important parts. We focused the conclusion on the main points of the article, making significant revisions. We believe that we have created a better structured and more informative discussion and conclusion. Once again, we express our gratitude for the constructive criticism and hope that we have achieved a satisfactory effect as a result of our editing.
- The extensive embryological discussion (lines 221-260) appears tangential to the systematic review's objectives
We carefully read the first part of the discussion, which included embryology, and significantly shortened this paragraph, thereby lightening the style of the article and focusing on the main issue of "anatomical features and consequences of surgical interventions." The first paragraph occupies part of lines 228-246. The co-authors are convinced that this improves the appearance of the article. Once again, we would like to thank the reviewer for their constructive criticism.
Once again, we, the co-authors, would like to thank the reviewer for their recommendations. We hope that with the edits we have made, our article will be worthy of acceptance in the Diagnostics journal and thus increase the readability of our scientific material.
With respect!
Reviewer 2 Report
Comments and Suggestions for Authors
This manuscript reports on anatomical aspect of the sphenoid sinus and its relationship with surgical strategy for pituitary surgery. A review of the published papers seems a good idea to maximize the value of these findings. Introduction is well written and contains the rational and aims of the study. Methods are well described and results well reported. At the beginning of the discussion I considera that the part referred to embryological development should be reduced as the goal of the manuscript is identification of anatomical features and implications for surgery. Aspect related to anatomy are well analyzed and also implications for surgical approach. In the section "implications for future research..." a sharable consideration are included but I suggest Authors to shorten this paragraph with only fundamental focus. In the same way the conclusions are too long, redundant and repetitive. Also this paragraph should be shortened. Figure 1 legend should be reduced as the data are reported in methods. In legend of figure 2 is useful to report number of cases evaluated to which the percentages refer. I have not other criticisms to report. In general the manuscript is interesting and well written.
Author Response
Dear Editors of Diagnostics journal,
Thank you for your edits to the article: Morphological analysis of the cavernous segment of the internal carotid artery: a retrospective, single-center study of its clinical significance.
The co-authors carefully reviewed the written material and tried to strike a balance between the two reviewers and to comply with the recommendations as much as possible in order to make the article even better and more useful for the readers of the journal.
Regarding the comments:
This manuscript reports on anatomical aspect of the sphenoid sinus and its relationship with surgical strategy for pituitary surgery. A review of the published papers seems a good idea to maximize the value of these findings. Introduction is well written and contains the rational and aims of the study. Methods are well described and results well reported. At the beginning of the discussion I considera that the part referred to embryological development should be reduced as the goal of the manuscript is identification of anatomical features and implications for surgery. Aspect related to anatomy are well analyzed and also implications for surgical approach. In the section "implications for future research..." a sharable consideration are included but I suggest Authors to shorten this paragraph with only fundamental focus. In the same way the conclusions are too long, redundant and repetitive. Also this paragraph should be shortened. Figure 1 legend should be reduced as the data are reported in methods. In legend of figure 2 is useful to report number of cases evaluated to which the percentages refer. I have not other criticisms to report. In general the manuscript is interesting and well written.
- The authors of the article appreciate the reviewer's positive assessment and will try to comply with the requirements as thoroughly as possible so that the article looks better and is useful to practicing physicians in the field when published. We have not made any changes to the abstract and introduction. We carefully read the first part of the discussion, which included embryology, and significantly shortened this paragraph, thereby lightening the style of the article and focusing on the main issue of "anatomical features and consequences of surgical interventions." The first paragraph occupies part of lines 228-246. The co-authors are convinced that this improves the appearance of the article. Once again, we would like to thank the reviewer for their constructive criticism.
- We thank the reviewer for the recommendation to shorten the section "Implications for future research in transsphenoidal surgery." We have read and analyzed the text and fully agree with the guidelines given that the text is too detailed and unnecessary. All this dilutes the main focus of the article, which is why we shortened the text, leaving the main points referring to the literature review. The new text covers paragraphs 454-477. We believe that this makes the paragraph more understandable and at the same time does not shift the focus of the topic.
- The conclusion section was reworked and shortened to cover the main focus of the article without unnecessarily diluting the information. We, the authors, believe that this has resulted in a good conclusion to the article, allowing readers to return to the main points of the article after reading the entire text. We would like to thank the reviewer for their recommendations, which we hope have significantly improved the quality of the text.
- We fully agree with the recommendations given to us for Figure 1 and the legend below it—the information has already been presented and is repeated, so we have shortened this section and left only the main focus of the diagram. We thank the reviewer for the recommendation.
- With regard to the recommendation given for Figure 2, the number of patients who are assessed as a percentage is indicated above in the text. We fully agree that there should be clarity about the number of patients to which the percentages refer, which is why we added in the text of the figure that it concerns 800 patients—this information is located on lines 185-186. We thank the reviewer for pointing out the small details in the article that would make it more appealing to readers.
Once again, we, the co-authors, would like to thank the reviewer for their recommendations. We hope that with the edits we have made, our article will be worthy of acceptance in the Diagnostics journal and thus increase the readability of our scientific material.
With respect!
Round 2
Reviewer 1 Report
Comments and Suggestions for Authors
The authors have addressed all of my comments. I have no additional comments.